# Characterization of Bulgarian Rosehip Oil by GC-MS, UV-VIS Spectroscopy, Colorimetry, FTIR Spectroscopy, and 3D Excitation–Emission Fluorescence Spectra

**DOI:** 10.3390/molecules30193964

**Published:** 2025-10-02

**Authors:** Krastena Nikolova, Tinko Eftimov, Natalina Panova, Veselin Vladev, Samia Fouzar, Kristian Nikolov

**Affiliations:** 1Faculty of Pharmacy, Medical University-Varna, Blvd Tzar Osvoboditel 84, 9000 Varna, Bulgaria; kr.nikolova@abv.bg (K.N.); nkpanova@gmail.com (N.P.); 2Centre de Recherche en Photonique, Département d’Informatique et d’Ingénierie, Université du Québec en Outaouais, 101 rue St-Jean Bosco, Gatineau, QC J8Y 3G5, Canada; fouzsam1977@gmail.com; 3Central Laboratory for Applied Physics, Bulgarian Academy of Sciences, 61 Blvd Sanct Peterburg, 4000 Plovdiv, Bulgaria; kristian.nikolov@clapbas.bg; 4Department of Mathematics, Physics and Information Technologies, Faculty of Economics, University of Food Technologies, 26 Maritsa Blvd., 4002 Plovdiv, Bulgaria; v_vladev@uft-plovdiv.bg

**Keywords:** *Rosa canina* L. oil, gas chromatography, colorimetry, UV-Vis spectrometry, FTIR, fluorescence 3D excitation-emission matrices, smartphone spectrometer

## Abstract

We report the study of seven commercially available rosehip oils (*Rosa canina* L.) using GC-MS, colorimetry (CIELab), UV-VIS, FTIR, and 3D EEM fluorescence spectroscopy, including using a smartphone spectrometer. GC-MS revealed two groups of oil samples with different chemical constituents: ω-6-dominant with 45–51% α-linolenic acid (samples S1, S2, and S5–S7) and ω-3-dominant with 47–49% α-linolenic, 7.3–19.1% oleic, 1.9–2.8% palmitic, 1.0–1.8% stearic, and 0.1–0.72% arachidic acid (S3, S4). In S1 PUFA content was found to be ~75% with ω-6/ω-3 ≈ 2:1. Favorable lipid indices of AI 0.0197–0.0302, TI 0.0208–0.0304, and h/H 33.0–50.6 were observed. The highest h/H (50.55) was observed in S5 and the lowest TI (0.0208) in S3. FTIR showed characteristic lines at ~3021, 2929/2853, 1749, and ~1370 cm^−1^, and PCA yielded 60–80% variation and separated S1 from the rest of the samples, while the clusters grouped S5 and S6. The smartphone spectrometer also reproduced the individual differences in sample volumes ≤ 1 µL under 355–395 nm UV excitation. The non-destructive optical markers reflect the fatty acid profile and allow fast low-cost identification and quality control. An integrated control method including routine optical screening, periodic CG-MS verification, and chemometric models to trace oxidation and counterfeiting is suggested.

## 1. Introduction

The rosehip (*Rosa canina* L.) is a species belonging to the Rosaceae family encompassing over 200 species and approximately 18,000 varieties worldwide. It thrives under a wide range of climatic conditions, including poor and infertile soils, and occurs naturally across Asia, the Caucasus, Europe, and Africa [1]. Vegetable oils are commonly used as ingredients in pharmaceuticals, dietary supplements, cosmetic products, and the food industry. In Europe, oils are often extracted from the seeds of rose species such as *Rosa eglanteria* L. and *Rosa canina*, both belonging to the Rosaceae family. The oil yield from these seeds is typically below 10% [2]. The oil exhibits antibacterial, anti-inflammatory, and antifungal properties and has been shown to inhibit the proliferation of cancer cells [3]. In addition to its rich fatty acid profile and the balanced ratio of ω-3 to ω-6 fatty acids which support regenerative processes in the skin layers, the oil is also abundant with *β*-carotene and vitamins C, A, B1, E, and K as well as with essential minerals such as potassium (K), calcium (Ca), sodium (Na), iron (Fe), and magnesium (Mg) [4].

Polyunsaturated fatty acids, known as essential fatty acids (ω-3 and ω-6), are not synthesized by the human body and must be obtained through a healthy diet. According to [5] examples of vegetable oils rich in polyunsaturated fatty acids include those derived from andiroba (*Carapa guianensis*), copaiba (*Copaifera langsdorffii*), almonds (*Prunus dulcis*), and rosehip (*Rosa aff rubiginosa*). During the processing of rosehip fruits, residual by-products from the seeds are generated, accounting for approximately 30 to 35% of the total fruit mass [6].

Due to their rich fatty acid composition, these seed residues can be further processed to obtain rosehip oil, contributing to a closed-loop production system and the development of zero-waste technologies. The resulting oil is increasingly utilized in the formulation of cosmetic and pharmaceutical products.

Although the extraction of oil from rosehip seeds is labor-intensive, in South America approximately 20,000 tons of seeds are processed annually to obtain oil [7], which finds application in cosmetics and medicine due to its content of transretinoic acid [8].

Due to its low extraction yield—typically only 4–7% of oil recovered from the dry weight of rosehip seeds depending on the method—rosehip oil is often adulterated with cheaper oils such as corn, soybean, or sunflower oil [9].

The use of multiple spectroscopic techniques to assess the authenticity of rosehip oil is necessary because its yield is low and the product is often adulterated with cheaper vegetable oils of similar chemical composition. The presence of diverse classes of bioactive compounds such as carotenoids, tocopherols, chlorophylls, and phenolics requires the application of different techniques. While GC-MS provides a reference chemical profile, complementary non-destructive spectroscopic methods (UV-Vis for pigment content, FTIR for functional groups, fluorescence for chlorophyll and oxidative markers, and colorimetry for visual parameters) allow rapid screening and quality differentiation, since the similarity or difference between samples cannot be established using a single method.

Therefore, the development of a set of optical, non-destructive, and rapid methods is of significant interest, with the aim of creating algorithms for distinguishing pure from adulterated oils. Although official methods and recommended practices established by the American Oil Chemists’ Society (AOCS) exist for assessing the quality of vegetable oils, they are time-consuming, require expensive reagents, generate environmental waste, and demand highly trained personnel who must strictly follow protocols to ensure analytical accuracy.

Numerous recommendations have been made for the implementation of non-destructive techniques as cost-effective alternatives to high-end analytical methods for differentiating pure oils from those adulterated with other plant-derived oils, especially those extracted from medicinal plants. Makky and Soni (2014), as well as Luna, Silva, Pinho, Ferré and Boqué (2013) recommend the use of UV-Vis spectroscopy as a suitable analytical technique [10,11]. Rohman, Riyanto, Sasi and Yusof [12] as well as Luna, Silva, Pinho, Ferré and Boqué [11] recommend mid-infrared (MIR) spectral analysis as a reliable technique for assessing the authenticity and quality of vegetable oils. Tankeu et al. [13] and Makky and Soni [10] recommend near-infrared (NIR) spectral analysis as an effective, rapid and non-destructive method for evaluating the authenticity and quality of vegetable oils [10,13]. Zahir, Saeed, Hameed, and Yousuf [14] employed Raman spectroscopy in their study of the physicochemical properties of edible oils and the evaluation of frying oil quality using Fourier transform infrared (FT-IR) spectroscopy. On the other hand, Silva, Filardi, Pepe, Chaves and Santos [15] utilized fluorimetry in combination with artificial neural networks for the classification of food vegetable oils.

The aim of the present study is to evaluate the effectiveness and applicability of non-destructive optical techniques such as infrared (IR) and UV-Vis spectroscopy, colorimetry, and fluorescence 3D excitation–emission matrices (EEMs) for the characterization and quality assessment of rosehip oils.

The main objective of the present study is not only to characterize the chemical composition of rosehip oils, but to also demonstrate the applicability of rapid and non-destructive optical methods (UV–VIS, FTIR, colorimetry, 3D EEM fluorescence spectroscopy, and smartphone-based spectroscopy) for quality screening. The observed correlations between fatty acid composition and optical parameters provide important evidence that such methods can be employed as practical tools for routine quality control and detection of adulteration in the market.

## 2. Results and Comments

### 2.1. GC-MS Analysis

#### 2.1.1. Summary of Results

The obtained gas chromatographic analysis results are presented in Table 1. The fatty acid content was calculated as mg/g lipid and expressed as relative percentage (%) of total fatty acids.

#### 2.1.2. Lipid Health Indices

To complete the analysis, the following lipid health indices were calculated based on the fatty acid profile of the rosehip oil samples by using Formulas (1)–(3) from Section 4.2.2: AI—atherogenic index, TI—thrombogenic index, h/H—hypo-/hypercholesterolemic ratio.

The results are listed in Table 2.

### 2.2. Transmission Spectroscopy

The transmittance spectra and their first derivatives obtained in the UV and visible ranges are presented in Figure 1a,b. As can be seen, the transmission is above 80% for wavelengths above 550 nm, with absorption bands around 612 nm and 672 nm. The low transmission under 500 nm in the blue-green suggests an absorption that can lead to meaningful fluorescence emission. Because of the evidently different transmission curves, the color characteristics should also differ and can be a source of analysis.

As the transmission spectra are individual we next consider the color measurement results.

### 2.3. Color Analysis

Using the equipment described in Section 4.2.3 and following the procedure, the seven samples were analyzed, and the results for their color parameters are summarized in Table 3.

The individual numbers from Table 1 clearly differentiate the samples since the error is less than the distance of the average value for a given parameter between the different samples.

### 2.4. FTIR Spectroscopy Results

In this study, the differences in the infrared spectra between the rosehip oil samples were evaluated and compared with the changes in their composition and structure. The IR spectra are presented in Figure 2.

In the IR spectra of rosehip oil several characteristic absorption bands are observed, associated with the following:Stretching vibrations of =C–H bonds around 3021 cm^−1^ which are indicative of the presence of unsaturated fatty acids.Absorption maxima at approximately 2929 cm^−1^ and 2853 cm^−1^ correspond to the asymmetric and symmetric stretching of –CH_2_ groups located in the aliphatic chains of fatty acids [2]. The intensity of these peaks is associated with the relative percentage of linoleic and linolenic acids.The peak around 1749 cm^−1^ is attributed to the stretching of the ester carbonyl (C=O) group in triglycerides [16]. This band is mainly linked to triglycerides and esters of polyunsaturated fatty acids.A peak near 1370 cm^−1^ is observed resulting from bending vibrations of CH_2_ and CH_3_ groups [17].

Normalization of the IR spectra of the studied rosehip oil samples was performed. Principal Component Analysis (PCA) was conducted, with PC1 explaining between 60% and 80% of the data variance, representing the intensity ratio between 1740 cm^−1^ and 2920 cm^−1^. PC2 accounts for an additional 10% to 20% of the variance and is associated with finer structural differences. particularly the intensity ratio at 1375/1465 cm^−1^ attributed to CH_3_/CH_2_ groups.

The robustness of the PCA model was validated by scree plot and cross-validation. As shown in Figure 3, PC_1_ accounted for 96.9% and PC_2_ for 3.1% of the variance, together explaining 100% of the data variability. The model demonstrated excellent fit and predictive ability, with R^2^ = 1.00 and Q^2^ = 1.00 (leave-one-out cross-validation). The PCA score plot confirmed the distinct grouping of the samples. Sample S1 remained clearly separated from the rest, while samples S2–S4 together with S7 formed a coherent cluster, and samples S5 and S6 consistently grouped into a separate cluster.

Three major groups were formed from the seven rosehip oil samples using hierarchical cluster analysis, as shown in Figure 4. Cluster 2 contained only S1, Cluster 3 contained S2 and S4, and Cluster 1 contained S3, S5, S6, and S7. This clustering closely matched the results of the PCA analysis. Hierarchical clustering displays both significant and minor differences and similarities, whereas PCA displays the variance structure in two dimensions. Given that the producer identified this sample as containing St. John’s Wort essential oil, it makes sense that S1 would clearly separate in both analyses. On the other hand, the grouping of S3, S5, S6, and S7 demonstrates how similar their compositions are. These findings demonstrate the effectiveness of FTIR and multivariate statistics in identifying and validating rosehip oil samples from various sources.

The FTIR results from Figure 2 permit the calculation of the IR peak ratios, which are indicative of the relative presence of ester groups compared to alkyl chains. These IR peaks ratios for the samples are summarized in Figure 5.

### 2.5. Three-Dimensional Excitation–Emission Matrix (EEM) Fluorescence Analysis

We finally consider the results from the fluorescence measurements as performed with an optical fiber spectrometer (Ocean Optics) and with smartphone spectrometers. Using the tunable monochromator we measured the fluorescence spectra of the seven samples with a fiber spectrometer and with a smartphone spectrometer for excitation wavelengths varying from 220 nm to 700 nm. The 3D plots are presented in Figure 6a1–b8.

Comparisons of the 3D plots in the left column (6a) and the right column (6b) reveal several important observations:(i)The 3D spectra of each sample are strictly individual, whether measured with a fiber spectrometer (left column) or with a smartphone spectrometer (right column). This observation implies that fluorescence spectra can be used to identify the particular type of oil.(ii)While the spectra are individual, the particular differences are in the spectral range of 400 nm (violet-blue) to 620 nm (orange-red). However, the spectra measured with the fiber spectrometer are weaker than those measured with the smartphone spectrometer, as is evident when we compare Figure 6a8,b8.(iii)The 3D spectra measured with a smartphone spectrometer appear drastically different from those measured with a fiber spectrometer, the most evident difference being the strong suppression of the blue (B) component at 400 nm to 475 nm, the boost in the middle green (G) component at 475 nm to 570 nm, and suppression of the red (R) component at 600 nm to 700 nm. These deformations of the spectra are in agreement with the R, G, B weight coefficients from Equation (6).(iv)The spectra taken with a smartphone usually exhibit, though to a different extent, a peak around 585 nm where the G and R filter transmissions intersect.

The reason for these peculiarities is the different spectral sensitivities of the spectrometer and the smartphone camera. The spectral sensitivity of the former monotonically increases in the range from 300 to 900 nm. On the other side, the transmission spectra of the smartphone camera are in the visible range from 400 nm to 700 nm, rather irregular, model-specific [18], and exhibit a maximum in the green around 550 nm. Also, the measurements with the spectrometer were taken at 1 s integration (exposure) time, while those with the Samsung smartphone were taken at 10 s, which is why the spectra are somewhat saturated.

To illustrate the model-dependent differences in the performance, comparisons of the spectra from three different smartphones are presented in Figure 7a1–c3, which show the following:(i)The model which best reproduces the 675 nm chlorophyll maximum is the Nothing Phone (R), while the model that exhibits the least spectral jump at 475 nm is the Xiaomi T11 Pro. The Samsung A41 is a compromise.(ii)In all of the phones the blue components from the photos of the spectra are as strong as the green components, and it is the weighting coefficients that introduce the discrepancies in the spectra in Figure 7b1–b3. In principle these deformations can be remedied by modifying the weight coefficients from Equation (6).

## 3. Discussion

### 3.1. Discussion on the CG-MS Results

The CG-MS results presented in Table 1 provide the possibility for a number of comments. In samples S1, S2, S5, S6, and S7, linoleic acid predominated (ranging from 45% to 51%), whereas in samples S3 and S4, α-linolenic acid was the major component (between 47% and 49%). The Bulgarian rosehip oils differ from those analyzed by [19], where arachidic acid predominated (32.93%), followed by linoleic acid (29.72%), heneicosanoic acid (19.27%), palmitoleic acid (7.02%), linolenic acid (4.20%), oleic acid (4.01%), and behenic acid (2.85%). In our study, oleic, linoleic, and α-linolenic acids were predominant, consistent with data reported for rosehip seed oils (*R. canina* L.) originating from different regions of the world [20,21]. Samples S3 and S4 exhibit a rich ω-3 profile, making them comparable to flaxseed oil. These samples might be suitable for the development of formulations with anti-inflammatory activity. Sample 1 shows high oxidative stability and a favorable ω-6/ω-3 ratio of 2:1. The high content (76%) of polyunsaturated fatty acids (PUFAs) suggests that the oil is suitable for nutraceutical applications, with antioxidant potential and favorable organoleptic properties. The lipid health indices from Table 2 suggest that sample S5 may be a candidate for the development of supplements intended to support cardiovascular health, due to its low AI and having the highest h/H ratio. Sample 3, with the lowest TI, is appropriate for the development of supplements aimed at reducing thrombotic risk. Sample 1, due to its balanced ω-3/ω-6 ratio and favorable values across all three lipid indices, suggests applications for the development of nutritional supplements.

No single internationally recognized standard for rosehip oil fatty acid composition exists. Nevertheless, a comparative analysis of the rosehip oils investigated in this study with other samples worldwide, reported in the literature based on the content of four predominant fatty acids, has been carried out. The results of the comparison are presented in Table 4.

The Bulgarian rosehip oil samples have the lowest palmitic acid content and resemble only the French samples. In terms of α-linolenic acid content, our samples are similar to those from Canada, Chile, and France. The Bulgarian samples contain lower levels of oleic acid, with some of them resembling those from Turkey, Hungary, and Canada.

### 3.2. Discussion on Optical Methods

#### 3.2.1. Transmission Spectra and Colorimetry

In the transmission spectra from Figure 3 the low transmittance coefficients in the UV range below 320 nm can be explained by the significant absorption of carotenoids and phenolic compounds. Similar observations have been reported by [28]. Absorption in the range of 325 to 350 nm is associated with the presence of phenols and flavonoids, respectively [29]. Transmittance maxima around 635 nm in cold-pressed oils as observed in our case have also been reported in [30].

The colorimetric parameters from Table 3 provide provide additional information on important correlations. Color intensity (C*) and redness (*a**) correlated with the presence of conjugated double bonds and pigments.

Sample S1 has the lowest value of chroma (C* = 130.3); i.e., it is the least saturated in color, correlating with its lower unsaturation level and pigment. It has the lowest value of the color parameter *a*, and the highest hue angle (*h*). S2 and S3 had similar color profiles, supporting the link between high α-linolenic acid and pigment retention, and showed the greatest similarity in their color parameters *a* and *b*.

S4 exhibits the highest lightness, while S6 and S7 have the highest values of the color parameter *a* (>35) and C* (>140), which determines the intense red-orange color and can potentially be explained by a higher content of β-carotene, consistent with their fatty acid profiles.

Samples S2, S3, S6, and S7 exhibit high *b* and *a* coefficients, indicating a dominant yellow-orange color. Similar results have been reported by [30,31], who note that when *b* > 130 and *a* > 25, the samples possess an intense color that correlates with carotenoid content. All samples from S2 to S7 fall within this range.

The relationship between fatty acid saturation and color intensity may reflect both compositional and oxidative stability factors.

This study demonstrates a clear correlation between fatty acid profile and optical characteristics of rosehip oil. Particularly, colorimetric values *a** and C* strongly correlate with higher PUFA content and serve as indicators of quality and authenticity.

#### 3.2.2. FTIR Spectroscopy

On the basis of the identified causes for the absorption bands and peaks we can conclude that FTIR spectroscopy can be used as a rapid qualitative tool for evaluating rosehip oils since it provides insight into structural differences in the fatty acid composition of the oils such as chain branching or degree of unsaturation.

With reference to Figure 3 the PCA clearly shows that sample S1 is widely separated from the rest, which may suggest elevated levels of polyunsaturated triglycerides or early-stage oxidation or be due to the extracts of Hipericum Perfuratum present in the sample. According to the hierarchical cluster analysis as illustrated in Figure 4, S1 is once again the most distant from all other clusters, which reinforces the results obtained from the PCA. Based on the dendrogram S5 and S6 form the closest pair, confirming their similar fatty acid compositions.

In Figure 5, sample 1 exhibits the highest ratios, which suggests a greater relative presence of ester groups compared to alkyl chains. This may reflect a higher concentration of linoleic and linolenic acid esters or an increased number of oxidized components.

#### 3.2.3. 3D EEM Fluorescence

In the observed spectra the peak around 675 nm is due to chlorophyll. As the RGB spectral range of the smartphones is up to 700 nm, this peak is better reproduced by a standard spectrometer, especially its tail above 700 nm, as the plots in Figure 6a8 show.

Of all the samples the strongest chlorophyll peak is in S5, followed by S7 and S6; it is negligible in S4 and non-detectable in S1, S2, and S3. This observation holds for both the fiber and the smartphone spectrometers. The peak is most efficiently excited around 405–410 nm. The intensity of the 675 nm peak relative to other peaks can be used as an identification parameter [32]. Its presence may be due to differences in production technology.

The peak around 450 nm, which is the strongest for sample S1 and the weakest for S5 and S7, is indicative of *β*-carotene. The peak is well pronounced in S2, S3, and S4 and less so in S6, but is a clear marker especially in the 3D plots measured with a standard spectrometer. The smartphone spectrometer clearly shows it on the photos of the spectra, but the low weighting coefficient for the blue component (B) from Equation (6) artificially suppresses it, though it is still highly visible. This means that for analytical purposes the weighting coefficients have to be changed. The peak is the most excitable at 330 nm excitation.

The higher intensity in the blue-green (500 nm) to green (520 nm) part of the spectrum is basically due to oxidation processes but may also indicate the presence of certain tocopherols. This maximum is the strongest in S1, followed by S2, S3, and S4, and it is very weak in the rest. The smartphone outlines a stronger emission in the blue-green (around 500 nm) for S1 at 360 nm excitation and for S5 and less so for S4, S3, and S2 at 320–330 nm excitation. With the exception of S5 all samples exhibited a higher intensity in the green for 360–370 nm excitation.

Excitations around 330 nm, 360 nm, and 410 nm thus cause maximum fluorescence at different spectral regions which are in the corresponding R, G, and B pixel transmission bands and reveal important information on the content of the samples.

The methods essentially differ by complexity, size, analysis duration, and cost. The GC-MS method, which provides the most detailed analysis, requires costly equipment, highly qualified personnel, complicated sample preparation, and at least 30 min per sample. The FTIR method requires costly equipment in a low-humidity environment and trained personnel to read the spectra, but the analysis is relatively fast and only requires small amounts of the sample. The UV-VIS and colorimetric analysis is fast and only requires small amounts of samples, but the equipment is stationary. The standard spectrometric 3D fluorescence analysis also requires costly stationary equipment and qualified personnel, and the typical sample volume is 1 mL (1 cm^3^). In our case, the use of an optical fiber spectrometer allowed the sample volume to be reduced to 1 μL (1 mm^3^). The smartphone spectrometer allows sub-μL volumes, and in combination with small size, low-cost UV LED excitation allows it to be a portable analytical instrument for fast identification analysis. The 3D EEM plots clearly show that ten UV LEDs emitting from 320 nm to 410 nm with a 10 nm spacing can reproduce the most important fluorescence characteristics of the samples, and a minimum of three UV LEDs at 320 nm, 360 nm, and 410 nm outline the most important emission bands.

Based on the above considerations, a combined control method can be proposed in the future which would include low cost, fast screening using smartphones in combination with UV LEDs to analyze both spectra and RGB color responses, followed by periodic tests with stationary equipment, and final CG-MS verification in combination with chemometric models for a more detailed identification, tracing oxidation and counterfeiting.

In view of the above, deeper future research that will reveal the correlation between the smartphone’s observable spectra and color characteristics, on the one hand, and the individual content of the oils under study, on the other, will be pursued. Also, more research is needed to make full use of the smartphone’s color characterization and transmission spectroscopic measurement capabilities and the correction of the individual transmission characteristics.

## 4. Materials, Methods, and Experimental Setups

### 4.1. Description of the Samples Under Study

The seven rosehip oil samples were purchased from the commercial market and were produced by Bulgarian manufacturers. In Table 5, data on the origin and method of obtaining the oils are presented.

Samples 1 and 5 are oils from *Rosa canina* L. extracted under a nitrogen atmosphere. Sample 5 is oil freshly extracted under a nitrogen atmosphere from organically grown rosehips, provided to us directly by the producer, prepared specifically for the purposes of this study and without subsequent refining or addition of antioxidants with a gel-stabilizing effect. Sample 1 is from a commercial sample from the same producer with the addition of St. John’s wort essential oil (*Hypericum perforatum*). The presence of St. John’s wort essential oil is indicated on the label, but the percentage ratio is not specified.

The lack of freshly extracted, non-commercial *Rosa canina* L. oil as a reference sample was compensated for by including sample S5 in the study, which was provided directly by the producer as rosehip oil ecologically produced under a nitrogen atmosphere, prepared specifically for the purposes of this study and without subsequent commercial refining.

### 4.2. Methods

#### 4.2.1. Gas Chromatographic Analysis

Sample preparation for the extraction of lipids (fatty acids) by GC-MS was carried out according to the methodology described by Ivayla Dincheva [33]. The fatty acid composition of the rosehip oil samples was determined using a gas chromatograph (Agilent GC 7890) equipped with a mass spectrometric detector (Agilent MSD 5975) and an HP-5MS column (30 m length, 0.32 mm internal diameter, 0.25 µm film thickness) (Agilent Technologies, Santa Clara, CA, USA). During the first two minutes the oven temperature was maintained at 100 °C. It was then increased to 180 °C at a rate of 15 °C min^−1^ and held for one minute. Finally, the temperature was ramped up to 300 °C at 5 °C min^−1^ and held for 10 min. The injector and detector temperatures were set at 250 °C and helium was used as the carrier gas at a flow rate of 1 mL min^−1^. Compound identification was performed by comparing retention times and Kováts retention indices with those of standard chemicals, as well as using mass spectral data from the NIST’08 library.

#### 4.2.2. Lipid Quality Indices

The atherogenic index (AI), thrombogenic index (TI), and hypocholesterolemic/hypercholesterolemic ratio were calculated using the equations suggested by Ulbricht and Southgate [34] and Santos-Silva et al. [35]:(1)$$AI=\frac{C_{12:0}+4.C_{14:0}+C_{16:0}}{MUFA+PUFA}$$(2)$$TI=\frac{C_{14:0}+C_{16:0}+C_{18:0}}{0.5\times \sum MUFA+0.5\times \sum \omega_{6}PUFA+3\times \sum \omega_{3}PUFA+\frac{\sum \omega_{3}PUFA}{\sum \omega_{6}PUFA}}$$(3)$$\frac{h}{H}=\frac{C_{18:1\omega_{9}}+\sum PUFA}{C_{14:0}+C_{16:0}}$$
where MUFA is monounsaturated fatty acids, and PUFA is polyunsaturated fatty acids.

#### 4.2.3. UV-VIS Spectra and Color Characteristics

The transmission spectra of the samples were analyzed using an Evolution Pro UV-Vis Spectrophotometer US USP/EP Standards Bundle (Thermo Fisher Scientific, Waltham, MA, USA) in the wavelength range from 250 nm to 750 nm employing a 1 cm quartz cuvette (≥1 mL) at room temperature (20 °C). The samples were poured directly into the cuvette and their spectral characteristics were recorded without dilution. The color parameters were calculated using VISIONlite ColorCalc software (version 5).

The color parameters of the oils used were determined in the CIELab colorimetric system after pre-tempering of the samples. The brightness (L) and color characteristics (*a* and *b*) in the CIELab colorimetric system were measured along with the color coordinates X, Y, and Z and chromaticity coordinates x and y in the XYZ colorimetric system under the following visual reference conditions: standard illuminant D65 (corresponding to a color temperature of 6504 K) and standard CIE 1964 observer (10° viewing angle). For the purposes of this study the CIELab system is more informative as it is designed for detecting small color differences. The dominant wavelength and color purity were also determined as well as parameters such as lightness (L), color saturation (C), and hue angle (*h*).(4)$$h_{ab}=arctg\left(\frac{b}{a}\right)$$(5)$$C=\sqrt{a^{2}+b^{2}}$$

#### 4.2.4. IR Spectra

We used a Jasco FT/IR-4X FTIR spectrophotometer to obtain Fourier transform infrared (FTIR) spectra (Jasco, Hachioji, Japan). The spectral range was from 4000 to 400 cm^−1^. To reduce background noise from the equipment each sample was scanned 32 times at a resolution of 2 cm^−1^. Each of the seven cold-pressed rosehip oil samples were scanned three times to ensure measurement repeatability and spectral consistency, and the averaged FT-IR spectrum is presented.

#### 4.2.5. Fluorescence Measurement Arrangements

(A)Standard 3D fluorescence spectral measurement setup

To measure the 3D excitation–emission spectra we used the experimental setup shown in Figure 8 in which a laser-driven white light source (Energetiq, Wilmington, MA, USA) was coupled to an Ocean Optics monochromator providing a tunable excitation in the 220 nm to 900 nm spectral range. The spectral measurements were performed using a standard fiber spectrometer (QE 65000, Ocean Optics, Duiven, The Netherlands).

The liquid of the samples was inserted into a glass capillary (∅1 mm/1.5 mm inner/outer diameter) which was then placed into a sample holder. The sample volume was about 1 μL (1 mm^3^). The SMA fiber connector tips of the excitation and receiving 800 μm quartz–polymer fibers were inserted at 90° with respect to each other, with the capillary with the sample being in the center. The emission spectrum was taken for each excitation wavelength, which was varied by the monochromator from 220 nm to 700 nm at a 10 nm increment. The 3D excitation emission spectrum was then plotted.

(B)Smartphone spectrometer measurement scheme

Apart from the standard fiber spectrometers, we also made use of smartphone spectrometers to analyze the fluorescence spectra since they are basically in the visible range. The smartphones used were Samsung Galaxy A51 (Samsung Electronics, Suwon-si, Republic of Korea), Xiaomi 11T Pro (Xiaomi, Beijing, China), and Nothing Phone Pro A069P (Nothing Technology Limited, London, UK). A 1000 L/mm transmission grating is placed in front of the smartphone’s camera, which is tilted to an angle to observe one of the diffraction maxima (Figure 9).

The 1.5 mm O.D. capillary with the oil samples is inserted along the X axis in a sample holder and is excited by a 3 mm UV LED along the Y axis. The fluorescence spectrum is observed along the Z axis through a 1 mm hole. The UV LEDs are 355 nm, 365 nm, 375 nm, 385 nm, and 395 nm. The photos of the spectra were processed using a Python-based program. The counts from each color pixel, R (red), G (green), and B (blue), were taken separately and then were converted to a grey scale (Y) whose counts were calculated in agreement with the International Commission on Illumination (CIE) as follows [32]:(6)Y=0.0722B+0.7152G+0.2126R

To calibrate the smartphone spectrometer, we used a reference fiber with which we launched some light from a 405 nm UV laser diode (LD) and a 633 nm He-Ne laser. The signals from the reference lasers were observed together with the fluorescence spectra.

#### 4.2.6. Statistical Methods

All measurements were performed in triplicate, and the results are presented as mean values ± standard deviation. Statistical comparisons among sample groups were performed using Duncan’s multiple range test, with significance set at *p* < 0.01. Data analyses were conducted using SPSS statistical software v23.0 (IBM Corp., Armonk, NY, USA).

To assess the similarity and variability among the analyzed rosehip oil samples using their IR spectra, three statistical methods were applied:✓Potential groupings.✓Hierarchical Principal Component Analysis (PCA): used to reduce dimensionality and visualize.✓Clustering: applied to identify sample similarity based on Euclidean distances.✓Peak Ratio Analysis: performed to evaluate relative differences in the intensities of functional groups.

## 5. Conclusions

Based on the detailed measurements performed using five different analytical methods, the following main conclusions can be formulated:

C1. Bulgarian rosehip oils are characterized by a high level of polyunsaturated fatty acids, and samples rich in ω-6 and ω-3 families are clearly distinguishable. The lipid health indices vary as follows: AI 0.0197–0.0302; TI 0.0208–0.0304; h/H 33.0–50.6. Sample S5, featuring the highest h/H ratio, is the most appropriate for products focused on cardioprotection, while S3, with the lowest TI, is best for formulas aimed at the reduction of the risk for thrombosis. Sample S1 is a balanced combination of ω-6/ω-3 ≈ 2:1 and ~76% PUFAs which can prospectively be used for nutraceutical applications.

C2. The optical characteristics and parameters (UV-Vis, CIELab, FTIR, fluorescent 3D EEM) correlate with the fatty acid profile and allow the quick differentiation of the samples. FTIR-PCE separates S1 from the rest. The chlorophyll fluorescence peak around 675 nm is the highest for sample S5, followed by S7 and S6, is non-detectable in S1 to S3, and is vanishingly small in S4. It can be used as an indicator of the presence of petal residue in the oil. The measurements with a smartphone-based hand-held spectrometer in combination with UV LEDs show that low-cost, portable screening equipment needing samples with microliter volumes is realistic and promising for field work. However, additional research is needed to validate it as a method.

C3. In view of the good correlations between the optical characteristics and the CG-MS data, we suggest a combined integrated protocol for the control of quality and authenticity, comprising low-cost routine optical screening and periodic higher-cost CG-MS analysis as well as a broadening of the set of samples to develop chemometric models for identification of samples, tracking oxidation, aging, and anti-counterfeiting.

An integrated approach, particularly the use of portable smartphone-based spectroscopy, will offer a practical application for rapid and low-cost quality control of rosehip oil. In future studies, we will optimize and validate the proposed methodology for assessing the quality of rosehip oil, supported by a database of freshly extracted oils obtained through physicochemical techniques.

## Figures and Tables

**Figure 1 molecules-30-03964-f001:**
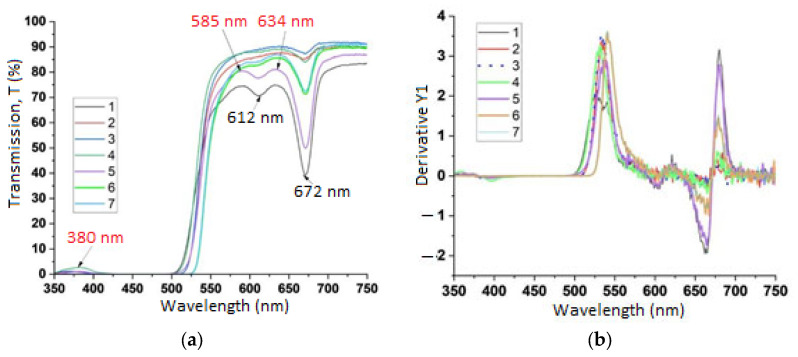
Transmission spectroscopic analysis: (**a**) transmittance spectra of rosehip oils in the UV and visible ranges; (**b**) first derivative of the transmittance spectra of rosehip oils.

**Figure 2 molecules-30-03964-f002:**
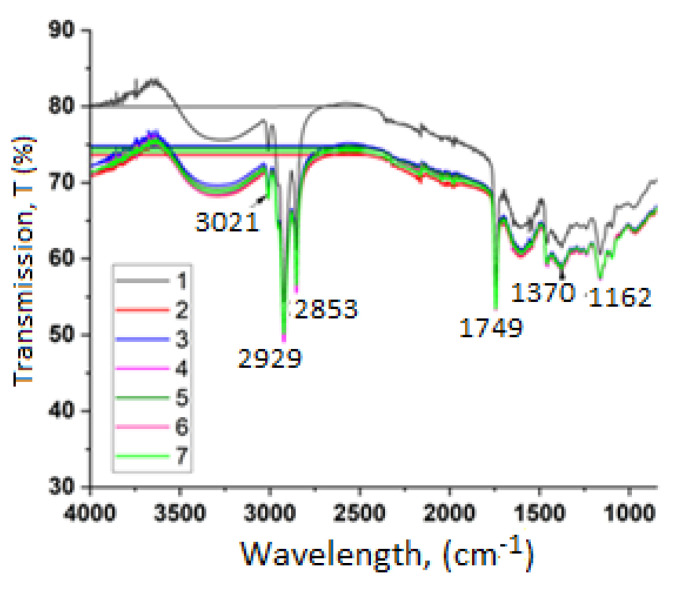
IR spectra for cold-pressed rosehip oils.

**Figure 3 molecules-30-03964-f003:**
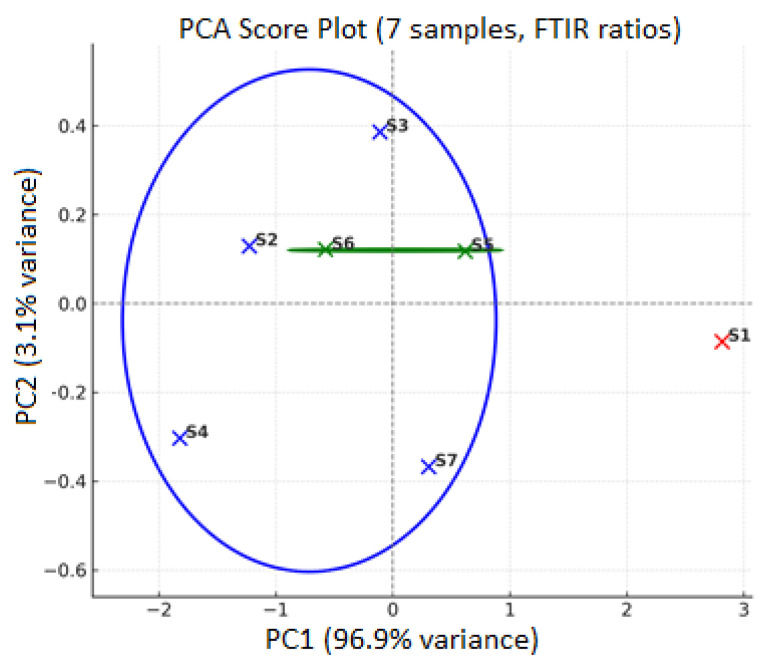
PCA plot showing distribution of rosehip oil samples based on normalized IR peak intensities.

**Figure 4 molecules-30-03964-f004:**
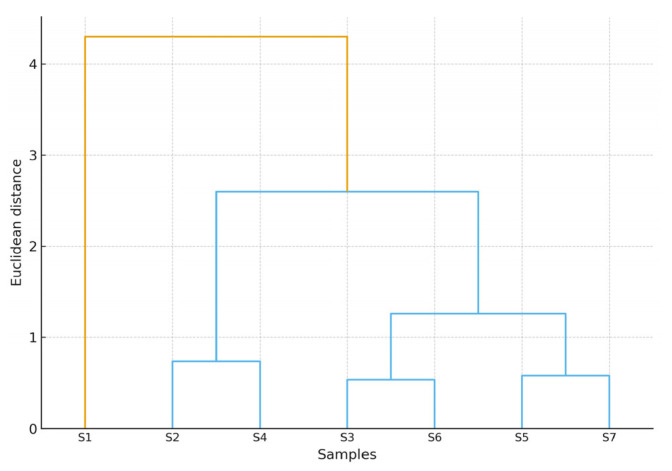
Dendrogram illustrating sample similarity based on IR spectra.

**Figure 5 molecules-30-03964-f005:**
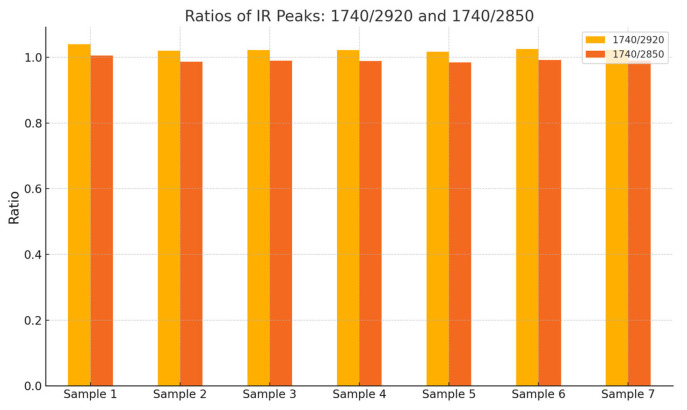
Ratio analysis between key IR peaks indicative of ester and alkyl content.

**Figure 6 molecules-30-03964-f006:**
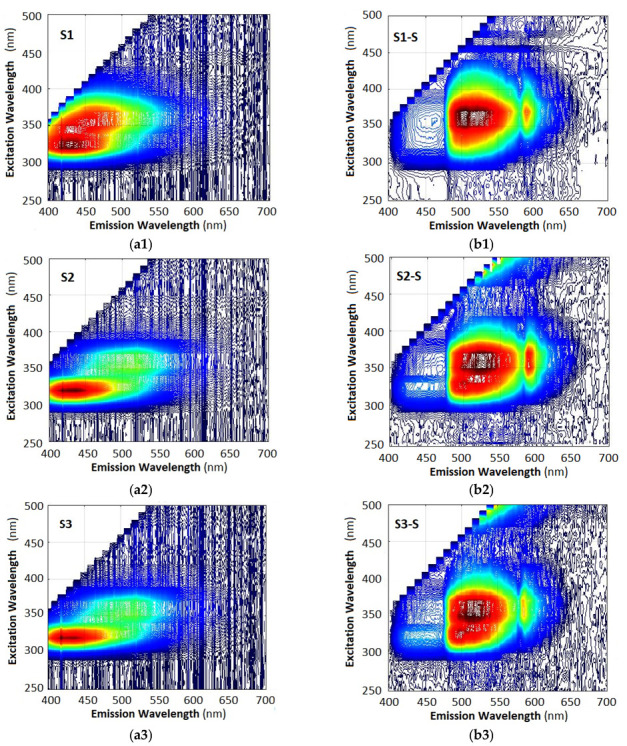
Topographic representation of the 3D excitation–emission spectra as measured with a fiber spectrometer—column (**a**), and with smartphone spectrometer—column (**b**): (**a1**,**b1**)—sample S1; (**a2**,**b2**)—S2; (**a3**,**b3**)—S3; (**a4**,**b4**)—S4; (**a5**,**b5**)—S5; (**a6,b6**)—S6; (**a7,b7**)—S7; (**a8**,**b8**) spectra of all the samples for 370 nm excitation.

**Figure 7 molecules-30-03964-f007:**
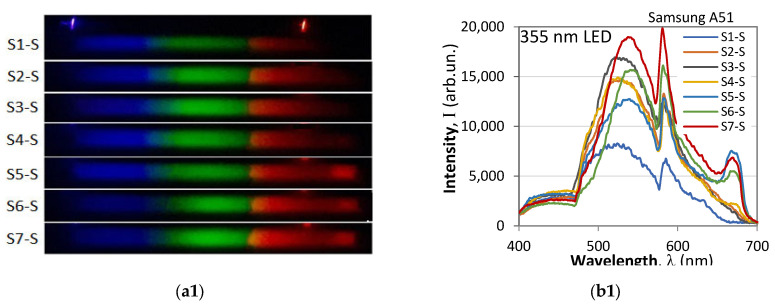
Fluorescence spectra of the seven samples taken with three different smartphones: (**a1**,**b1**) Samsung A51 (10 s exposure); (**a2**,**b2**) Nothing Phone (R) (25 s exposure); (**a3**,**b3**) Xiaomi 12T Pro (25 s exposure).

**Figure 8 molecules-30-03964-f008:**
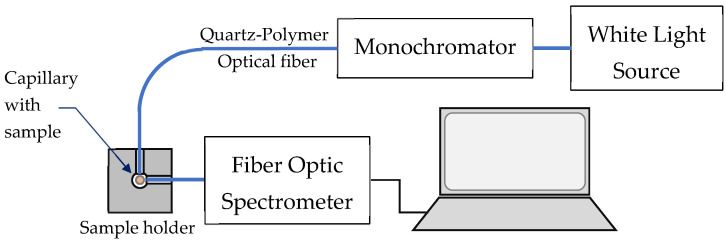
Experimental setup used to measure the 3D excitation–emission matrices, making use of a white light source, a monochromator, and a fiber optic spectrometer.

**Figure 9 molecules-30-03964-f009:**
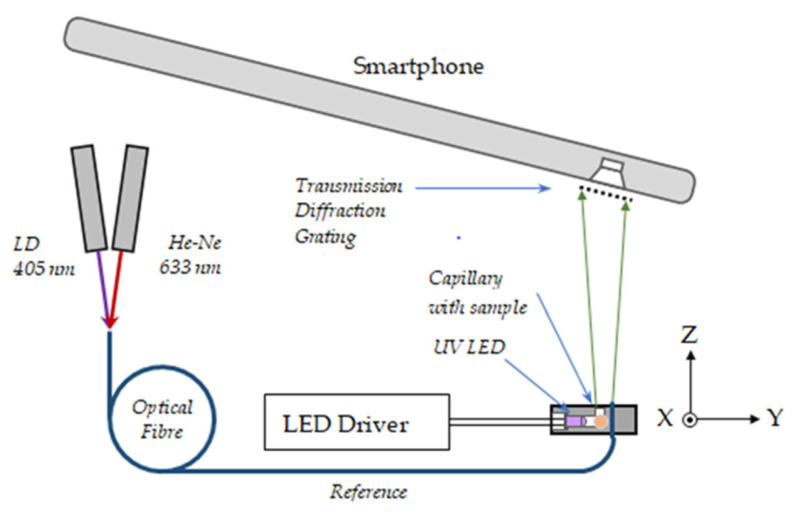
Experimental setup used to measure the fluorescence spectra using a smartphone spectrometer with 405 nm and 633 nm laser calibration.

**Table 1 molecules-30-03964-t001:** Fatty acid composition of rosehip oils.

Fatty Acid, Methyl Ester	Molecular Formula	RT	RI	Content. %						
				1	2	3	4	5	6	7
				Mean ± SD	Mean ± SD	Mean ± SD	Mean ± SD	Mean ± SD	Mean ± SD	Mean ± SD
Palmitic acid C16:0	C_16_H_32_O_2_	26.95	1920	2.2 ± 0.09 ^c^	2.77 ± 0.12 ^a^	2.52 ± 0.12 ^b^	2.8 ± 0.12 ^a^	1.9 ± 0.08 ^d^	2.06±0.10 ^d^	2.00 ± 0.10 ^d^
γ-linolenic acid C18:3	C_18_H_30_O_2_	29.76	2055	0.10 ± 0.00 ^d^	0.15 ± 0.01 ^b^	0.16 ± 0.01 ^b^	0.1 ± 0.01 ^c^	nd	0.22 ± 0.01 ^a^	nd
Linoleic acid Cl8:2	C_18_H_32_O_2_	30.30	2081	50.0 ± 2.10 ^a^	45.00 ± 1.98 ^b^	30.10 ± 0.44 ^d^	35.05 ± 1.40 ^c^	49 ± 2.13 ^a^	51.02 ± 2.45 ^a^	50.54 ± 0.48 ^a^
Oleic acid C18:1	C_18_H_34_O_2_	30.34	2090	19.1 ± 0.81 ^a^	10.12 ± 0.45 ^e^	16.03 ± 0.77 ^b^	12.16 ± 0.49 ^d^	13.29 ± 0.57 ^c^	7.31 ± 0.35 ^g^	8.33 ± 0.41 ^f^
α-Linolenic acid C18:3	C_18_H_30_O_2_	30.45	2098	26.1 ± 1.10 ^e^	39.50 ± 1.74 ^b^	49.20 ± 2.36 ^a^	48.20 ± 1.93 ^a^	33.62 ± 1.45 ^d^	36.56 ± 1.75 ^c^	37.00 ± 1.83 ^c,b^
Stearic acid C18:0	C_18_H_36_O_2_	30.75	2123	1.16 ± 0.05 ^d^	1.7 ± 0.07 ^a,b^	1.07 ± 0.05 ^e^	1.29 ± 0.05 ^c^	1.11 ± 0.05 ^e^	1.83 ± 0.09 ^a^	1.00 ± 0.05 ^f^
Gadoleic acid C20:1	C_20_H_38_O_2_	33.60	2220	0.44 ± 0.02 ^a^	0.35 ± 0.02 ^b^	0.31 ± 0.01 ^c^	0.18 ± 0.01 ^e^	0.16 ± 0.01 ^e^	0.40 ± 0.02 ^a^	0.21 ± 0.01 ^d^
9-cis-Eicosenoic acid C21:1	C_21_H_40_O_2_	33.72	2287	nd	0.25 ± 0.01 ^a^	nd	nd	nd	0.14 ± 0.01 ^b^	nd
Arachidic acid C20:0	C_20_H_40_O_2_	34.17	2333	0.72 ± 0.03 ^a^	0.16 ± 0.01 ^d^	0.28 ± 0.01 ^c^	0.10 ± 0.00 ^e^	0.12 ± 0.01 ^e^	0.29 ± 0.01 ^c^	0.36 ± 0.02 ^b^
Behenic acid C22:0	C_23_H_46_O_2_	37.40	2525	nd	nd	0.34 ± 0.02 ^a^	nd	0.13 ± 0.01 ^c^	0.17 ± 0.01 ^b^	0.15 ± 0.01 ^c^

Means in a row with a common superscript letter (a–g) differ (*p* < 0.05) as analyzed by Duncan’s test.

**Table 2 molecules-30-03964-t002:** Lipid health indices for Bulgarian rosehip oil.

	AI (Atherogenic Index)	TI (Thrombogenic Index)	h/H (Hypo-/Hyper Cholesterolemic Ratio)
S1	0.0235	0.0300	42.36
S2	0.0291	0.0304	34.16
S3	0.0263	0.0208	37.83
S4	0.0302	0.0246	33.01
S5	0.0197	0.0227	50.55
S6	0.0216	0.0278	46.06
S7	0.0207	0.0210	48.15

**Table 3 molecules-30-03964-t003:** Color characteristics of rosehip oils from Bulgaria.

Sample	S1	S2	S3	S4	S5	S6	S7
X	73.21 ± 1.02 ^e^	86.11 ± 1.74 ^b^	89.05 ± 2.03 ^a^	89.42 ± 1.81 ^a^	78.91 ± 1.03 ^d^	80.47 ± 1.23 ^c^	81.93 ± 1.22 ^c^
Y	56.64 ± 0.98 ^e^	63.86 ± 0.76 ^c^	66.22 ± 0.55 ^b^	68.53 ± 0.65 ^a^	58.94 ± 0.34 ^d^	56.30 ± 0.78 ^e^	57.48 ± 0.99 ^d^
Z	0.21 ± 0.05 ^a^	0.13 ± 0.01 ^b^	0.14 ± 0.02 ^b^	0.24 ± 0.03 ^a^	0.14 ± 0.03 ^b^	0.05 ± 0.01 ^c^	0.05 ± 0.00 ^c^
x	0.563 ± 0.01 ^c^	0.574 ± 0.002 ^b^	0.573 ± 0.003 ^b^	0.565 ± 0.002	0.572 ± 0.001 ^b^	0.588 ± 0.001 ^a^	0.578 ± 5 × 10^−3 b^
y	0.435 ± 0.001 ^a^	0.425 ± 0.002 ^b^	0.426 ± 0.006 ^b^	0.433 ± 0.004 ^a^	0.427 ± 0.003 ^b^	0.412 ± 0.008 ^c^	0.412 ± 0.003 ^c^
L	79.98 ± 1.22 ^d^	83.89 ± 1.54 ^b^	85.11 ± 1.32 ^a^	86.27 ± 1.65 ^a^	81.26 ± 1.22 ^c^	79.79 ± 1.34 ^d^	80.45 ± 1.11 ^d^
*a*	21.36 ± 0.76 ^e^	28.66 ± 0.34 ^b^	28.59 ± 0.21 ^b^	24.22 ± 0.24 ^d^	26.83 ± 0.12 ^c^	36.11 ± 0.14 ^a^	35.93 ± 0.11 ^a^
*b*	128.56 ± 2.12 ^f^	138.79 ± 2.15 ^a,b^	140.60 ± 2.71 ^a^	138.27 ± 1.99 ^a,b^	134.04 ± 1.76 ^d,e^	135.52 ± 1.56 ^d^	136.55 ± 1.65 ^b,c^
C	130.31 ± 0.98 ^e^	141.72 ± 0.76 ^b^	143.47 ± 0.45 ^a^	140.37 ± 1.03 ^c^	136.70 ± 1.11 ^d^	140.24 ± 1.08 ^c^	141.20 ± 1.17 ^b^
h	80.56 ± 2.56 ^a^	78.33 ± 2.34 ^b^	78.51 ± 1.97 ^b^	80.06 ± 1.34 ^a^	78.68 ± 1.56 ^b^	75.08 ± 1.38 ^c^	75.26 ± 1.98 ^c^

Means in a column with a common superscript letter (a–f) differ (*p* < 0.05), as analyzed by Duncan’s test.

**Table 4 molecules-30-03964-t004:** Comparative fatty acid compositions of rosehip oils from different regions of the world.

Fatty Acid, %	Country, References								
	Poland	Germany	France	China	Chile	Canada	Turkey	Hungary	Bulgaria
	[22]	[23]	[24]	[25]	[26]	[27]	[1,21]	[18]	Our Results
Palmitic acid C16:0	4.2–4.8	3.1	0–4.68	4	3.33–4.97	3.70	-	3.60–7.87	1.90–2.80
Linoleic acid Cl8:2	44.4–51.7	36.7	47.02–50.25	56.5	42.2–47.9	37.10	51.1–54.05	35.94–54.75	35.50–51.02
Oleic acid C18:1	14.7–16.3	18.8	-	34.2	12.4–14.8	19.70	19.3–19.5	16.25–22.11	7.31–19.1
α-Linolenic acid C18:3	21.5–31.8	14.3	33.02–40.21	1.7	28.4–31.1	30.75	19.4–21.4	20.29–26.48	26.1–49.2

**Table 5 molecules-30-03964-t005:** Origins of investigated Bulgarian rosehip oils.

	Method of Obtainment	Region	Type
S1	Extracted under a nitrogen atmosphere	Kazanlak, Bulgaria	*Rosa canina* L.
S2	Cold-pressed	Central Bulgaria	wild rosehips, petrified, of type *Rosa canina* L.
S3	Cold-pressed	Central Bulgaria	wild rosehips, petrified, of type *Rosa canina* L.
S4	Cold-pressed	Danubian Plain	*Rosa canina* L.
S5	Extracted under a nitrogen atmosphere	Kazanlak, Bulgaria	*Rosa canina* L.
S6	Cold-pressed	Southern Bulgaria	*Rosa canina* L.
S7	Cold-pressed	Southern Bulgaria	*Rosa canina* L.

## Data Availability

The original contributions presented in this study are included in the article. Further inquiries can be directed to the corresponding author.

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
