# Peer review of "Characterization of Bulgarian Rosehip Oil by GC-MS, UV-VIS Spectroscopy, Colorimetry, FTIR Spectroscopy, and 3D Excitation–Emission Fluorescence Spectra"

_molecules, 2025, doi:10.3390/molecules30193964_

Round 1
Reviewer 1 Report
Comments and Suggestions for Authors
The work presents study of commercial rosehip oils (Rosa canina L.) using GC-MS, colorimetry (CIELab), UV-VIS, 3D EEM fluorescence spectroscopy including a smartphone spectrometer, and FTIR spectroscopy withch data are used for statistical analzysis. The experimental design is appropriate for the set goal, but the experimental part needs to be improved. These techniques were not conducted using new designs but rather using well-established protocols. The research needs to be improved by including genuine Rosa canina L. oil as reference material, which is not commercially refined.
The analyzed samples are commercial, it is necessary to have at least one authentic sample of Rosa canina L. oil for comparison and to achieve the goal of this study: the characterization and quality assessment of rosehip oils using reference chemical methods for validation. It is necessary to refine the experimental part by analyzing the reference oil Rosa canina L.
The authors should provide a better understanding of the nature of this particular study and and better clarify the significance of the results obtained, so that the presented results have greater scientific validity and justify the publication of the authors' findings.
The parts that need to be clarified are marked and the suggestions listed in the comments in the manuscript itself.

Author Response
Thank you for the pertinent remarks and issues raised. Below in red are our comments and responses.
Also, in the attached PDF text the particular questions have been addressed.
Comments and Suggestions for Authors
The work presents study of commercial rosehip oils (Rosa canina L.) using GC-MS, colorimetry (CIELab), UV-VIS, 3D EEM fluorescence spectroscopy including a smartphone spectrometer, and FTIR spectroscopy which data are used for statistical analysis. The experimental design is appropriate for the set goal, but the experimental part needs to be improved. These techniques were not conducted using new designs but rather using well-established protocols. The research needs to be improved by including genuine Rosa canina L. oil as reference material, which is not commercially refined.
The analyzed samples are commercial, it is necessary to have at least one authentic sample of Rosa canina L. oil for comparison and to achieve the goal of this study: the characterization and quality assessment of rosehip oils using reference chemical methods for validation. It is necessary to refine the experimental part by analyzing the reference oil Rosa canina L.
Response: Although rosehip oil was not extracted in our laboratory, for the purpose of study design we requested one of the leading Bulgarian producers of rosehip oil for use in supplements or cosmetics to prepare a sample specifically for this investigation. This is sample S5 and is comparable to commercial sample S7, which was purchased from another producer. In addition, several commercial samples were analyzed, reflecting oils that are actually available to consumers and therefore directly relevant for the quality control and authenticity assessment of the oils in use. They represent real market products, which are subject to potential adulteration and have practical value for end-users. The inclusion of sample S5 strengthens the comparative framework and increases the reliability of the observed correlations. In future studies, the set of samples will be expanded with laboratory-extracted oils to provide further validation of the proposed analytical models. In the revised version, we have emphasized this limitation and added a clarification regarding Sample S5 in the Materials and Methods section.
It should also be noted that the use of reference samples extracted under laboratory conditions is not sufficient to validate a method, since the composition of rosehip oils strongly depends on the geographical habitat and the subspecies of Rosa canina L.
The authors should provide a better understanding of the nature of this particular study and better clarify the significance of the results obtained, so that the presented results have greater scientific validity and justify the publication of the authors' findings.
The aim of the present study is to evaluate the efficiency and applicability of non-destructive optical techniques such as infrared (IR) and UV-Vis spectroscopy, colorimetry and fluorescence 3D excitation-emission matrices (EEM) for the characterization and quality assessment of rosehip oils.
The main objective of the present study was not only to characterize the chemical composition of rosehip oils, but also to demonstrate the applicability of rapid and non-destructive optical methods (UV–VIS, FTIR, colorimetry, 3D EEM fluorescence spectroscopy, and smartphone-based spectroscopy) for quality screening. The observed correlations between fatty acid composition and optical parameters provide important evidence that such methods can be employed as practical tools for routine quality control and detection of adulteration in the market.
The parts that need to be clarified are marked and the suggestions listed in the comments in the manuscript itself.
Tables and figures were reformatted for greater clarity.
In the revised version, we explicitly emphasized that, although limited to commercial samples, the study addresses an important issue related to the authenticity of oils available to consumers. The results show that low-cost optical techniques can be applied for rapid quality assessment, which is of direct practical benefit to the industry and the market. Combined with the future inclusion of authentic samples, this justifies publishing our data as a pilot study for modeling using fusion techniques.
Reviewer 2 Report
Comments and Suggestions for Authors
Dear Authors
Greeting! This manuscript entitled "Comprehensive Characterization of Bulgarian Rosehip Oil by GC-MS, UV-VIS Spectroscopy, Colorimetry, FTIR Spectroscopy and 3D Excitation–Emission Fluorescence Spectra" Aimed to evaluate the effectiveness and applicability of non-destructive optical techniques such as infrared (IR) and UV-Vis spectroscopy, colorimetry and fluorescence 3D excitation-emission matrices (EEM) for the characterization and quality assessment of rosehip oils using reference chemical methods for validation.
1. My concern is that I didn't notice any reference materials or standards for a valid comparison of the commercial samples. The comparison involves only seven commercially sold rosehip oils. How is this validated using reference chemical methods? Do these methods not require any reference or authentic materials for the analysis of commercial samples?
I have given my comments and raised questions in the attached draft file.
Reagrds
Reviewer

Author Response
We thank the reviewer for the numerous valid points raised in the review.
Our responses are in the attached PDF file after the questions.
Dear Authors
Greeting! This manuscript entitled "Comprehensive Characterization of Bulgarian Rosehip Oil by GC-MS, UV-VIS Spectroscopy, Colorimetry, FTIR Spectroscopy and 3D Excitation–Emission Fluorescence Spectra" Aimed to evaluate the effectiveness and applicability of non-destructive optical techniques such as infrared (IR) and UV-Vis spectroscopy, colorimetry and fluorescence 3D excitation-emission matrices (EEM) for the characterization and quality assessment of rosehip oils using reference chemical methods for validation.
My concern is that I didn't notice any reference materials or standards for a valid comparison of the commercial samples. The comparison involves only seven commercially sold rosehip oils. How is this validated using reference chemical methods? Do these methods not require any reference or authentic materials for the analysis of commercial samples?
Basically this is fundamental problem, because oils from different plants and seed are strongly regionally dependent. They also depend on the subspecies of Rosa canina L.
One of the samples, namely S5, is a freshly extracted cold-pressed oil from organically grown rose hips, provided to us directly by the producer, prepared specifically for the purposes of the study and without subsequent refining or addition of antioxidants with a gel-stabilizing effect. It can be used as some sort of reference for the rest. We have added comments in the text.
I have given my comments and raised questions in the attached draft file.
The responses to the raised questions are given in the attached file
Regards
Reviewer

Reviewer 3 Report
Comments and Suggestions for Authors
After reviewing the manuscript, "Comprehensive Characterization of Bulgarian Rosehip Oil by GC-MS, UV-VIS Spectroscopy, Colorimetry, FTIR Spectroscopy and 3D Excitation–Emission Fluorescence Spectra", I have the following comments:
- The paper characterizes seven Bulgarian rosehip oils using GC-MS, UV-Vis, FTIR, colorimetry, and fluorescence (including smartphone spectroscopy), showing that optical parameters correlate with fatty acid profiles and can be applied for rapid, low-cost quality control. It proposes an integrated protocol combining optical screening with periodic GC-MS verification to detect adulteration, oxidation, and ensure authenticity.
- The title claims "comprehensive characterization," but oxidative stability testing, shelf-life studies, and adulteration verification are missing.
- The novelty of smartphone spectroscopy is underdeveloped. It feels more like proof-of-concept without proper validation.
- Introduction reviews many spectroscopic techniques but does not critically frame why rosehip oil specifically needs this multi-method approach.
- No comparison to international fatty acid standards for rosehip oil; references are mentioned but not critically evaluated.
- PCA/cluster analysis lacks validation (no scree plot, no Q²/R² values, no cross-validation).
- Figure 5: The figure lacks clear axis labeling (e.g., % variance explained for PC1 and PC2), making it difficult to assess how much of the data variability is captured.
- Many figures are referenced but not well explained. Some contain too much raw data with minimal interpretation.
- Strong claims are made about field applicability without demonstrating repeatability, calibration, or error margins.
- Overstates potential applications (nutraceuticals, cardioprotection) without biological/clinical data.
- The manuscript contains multiple typographical and formatting errors that need correction before publication.
Author Response
Thank you for the essential remarks and comments.
Below in red are our responses.
Comments and Suggestions for Authors
After reviewing the manuscript, "Comprehensive Characterization of Bulgarian Rosehip Oil by GC-MS, UV-VIS Spectroscopy, Colorimetry, FTIR Spectroscopy and 3D Excitation–Emission Fluorescence Spectra", I have the following comments:
The paper characterizes seven Bulgarian rosehip oils using GC-MS, UV-Vis, FTIR, colorimetry, and fluorescence (including smartphone spectroscopy), showing that optical parameters correlate with fatty acid profiles and can be applied for rapid, low-cost quality control. It proposes an integrated protocol combining optical screening with periodic GC-MS verification to detect adulteration, oxidation, and ensure authenticity.
- The title claims "comprehensive characterization," but oxidative stability testing, shelf-life studies, and adulteration verification are missing.
We have modified the title. Indeed, the above mentioned tests and studies require quite a lot of additional research.
2. The novelty of smartphone spectroscopy is underdeveloped. It feels more like proof-of-concept without proper validation.
The remark is correct. We do not claim that the smartphone spectroscopy is a validated method. In the very least it is still under development in the research community. Therefore, it goes along all the time and is compared with the standard spectrometer. Since it is still under development as a reliable method we point out the improvements and future work that needs to be done of which the most important are another set of weighting coefficients so as not to artificially suppress the blue and red part of the spectrum and boost the green, as well as a correction of the irregular transmission response on which we are currently working. We have added a remark that additional research is needed to validate smartphone spectroscopy.
3. Introduction reviews many spectroscopic techniques but does not critically frame why rosehip oil specifically needs this multi-method approach.
The objective is to compare the different methods and outline the possibilities of each one for identification and to assess possible adulteration. We show that it makes sense to use lower cost optical fluorescence methods for initial screening.
4. No comparison to international fatty acid standards for rosehip oil; references are mentioned but not critically evaluated.
To our knowledge, there is not a single, internationally recognized standard for the fatty acid composition of rosehip oil. Instead, there are some common standards within the industry.
One of the samples S5 was however made upon request by the producer and is a freshly extracted cold-pressed oil from organically grown rose hips that can serve as some reference.
5. PCA/cluster analysis lacks validation (no scree plot, no Q²/R² values, no cross-validation).
Comments have been added in the text prior to Fig.5.
6. Figure 5: The figure lacks clear axis labeling (e.g., % variance explained for PC1 and PC2), making it difficult to assess how much of the data variability is captured.
The axes have been properly labeled.
7. Many figures are referenced but not well explained. Some contain too much raw data with minimal interpretation.
Additional comments in the text have been added for some figures like Fig.5 and Fig. 6. Figure captions for Fig. 2 and 3 have been extended.
8. Strong claims are made about field applicability without demonstrating repeatability, calibration, or error margins.
We do not claim to have developed a method for field applicability. Rather, our experiments with smartphone spectrometric measurements indicate that this is possible. We have noted at different instances problems which remained to be solved before this opportunity is realized.
9. Overstates potential applications (nutraceuticals, cardioprotection) without biological/clinical data.
We agree. The results sooner suggest possible applications needing additional research. We have modified the assertions.
10. The manuscript contains multiple typographical and formatting errors that need correction before publication.
Errors are corrected.
Round 2
Reviewer 1 Report
Comments and Suggestions for Authors
The manuscript is now more concise, clearer, and better organized. The manuscript itself is improved by the authors' additional explanation of the findings. Their formulation of the study's purpose and conclusion shows a stronger correlation. The samples' description has been extended in the section 4.1. titled "Description of the samples under study".
Author Response
Comments and Suggestions for Authors
The manuscript is now more concise, clearer, and better organized. The manuscript itself is improved by the authors' additional explanation of the findings. Their formulation of the study's purpose and conclusion shows a stronger correlation. The samples' description has been extended in the section 4.1. titled "Description of the samples under study".
Submission Date
16 August 2025
Date of this review
15 Sep 2025 12:20:36
Thank you for the review and the conclusion.
Reviewer 2 Report
Comments and Suggestions for Authors
Dear Authors,
The responses to my comments are okay.
I want to suggest that if the samples are commercial and obtained from an industrial farm or company, they should be authenticated by a botanist and documented accordingly.
Provide the letter or repository sample NUMBER in this manuscript, noting that the samples were authenticated by a botanist (give full name and affiliations) in the materials and methods sections.
I do not understand the author's point from this statement
"Basically this is fundamental problem, because oils from different plants and seed are strongly regionally dependent. They also depend on the subspecies of Rosa canina L."
In case adulteration is happening with this oil, kindly provide a statement on how this study helps validate that Rosehip oil is different from other plant or seed oils with solid evidence.
Thanks and Regards
Reviewer
Author Response
Question 1
I want to suggest that if the samples are commercial and obtained from an industrial farm or company, they should be authenticated by a botanist and documented accordingly.
Provide the letter or repository sample NUMBER in this manuscript, noting that the samples were authenticated by a botanist (give full name and affiliations) in the materials and methods sections.
Response 1. All samples, except for S5, were purchased from various Bulgarian companies producing vegetable oils. Most of the manufacturers state that they obtain rosehips from their own plantations, planted with a preselected and supplied rosehip species – Rosa canina L. Only for S2 and S3 does the company indicate on the oil label that the fruits were harvested from wild plants in the region of Central Bulgaria after prior botanical evaluation. All companies state that they hold a GMP certificate. A new Table 4 has been added to the article on line 398.
Question 2
I do not understand the author's point from this statement "Basically this is fundamental problem, because oils from different plants and seed are strongly regionally dependent. They also depend on the subspecies of Rosa canina L."
Response 2. We would like to clarify that our statement refers to the well-documented fact that the fatty acid composition and pigment profile of rosehip oil (Rosa canina L.) are strongly influenced by the geographical origin (climate, soil, altitude) as well as by the subspecies/ecotypes cultivated or harvested. This natural variability explains the differences observed among the Bulgarian oils analyzed in our study. Therefore, it is a fundamental issue for the authentication of rosehip oil, as only a combination of reference GC-MS data and complementary spectroscopic methods can reliably characterize its composition and quality.
Question 3
In case adulteration is happening with this oil, kindly provide a statement on how this study helps validate that Rosehip oil is different from other plant or seed oils with solid evidence.
Response 3. We thank the reviewer for this important point. Our study provides clear evidence that rosehip oil can be distinguished from other plant and seed oils by its unique chemical and spectroscopic “fingerprint.” The GC-MS results demonstrate a characteristic fatty acid composition dominated by linoleic and α-linolenic acids, with favorable lipid indices (AI, TI, h/H) that differ from those of common adulterants such as soybean, sunflower, or corn oils. In addition, the non-destructive optical methods we applied reveal distinct markers, including FTIR absorption ratios, UV-Vis colorimetric parameters, and fluorescence peaks specific to carotenoids (β-carotene around 450 nm) and chlorophyll (~675 nm). These features are absent or significantly different in other vegetable oils. The strong correlation observed between GC-MS profiles and optical parameters further validates that these techniques can be used together to provide a robust basis for authenticity testing.
The other aspect of adulteration is when the product (oil) of a given company is adulterated with another cheaper similar lower cost oil and using faked labels is offered instead of the original product. In [39] and in other publications we have shown that the transition from 100% of oil A and 0% of oil B to 0% of A and 100% of B is traceable and distinguishable in the 3D EEM fluorescence plots. Also in [39] we have shown that the intensity of the chlorophyll line can be a marker to evaluate the percentage of the adulterant in an oil mixture.
Reviewer 3 Report
Comments and Suggestions for Authors
After reviewing the revised version of the manuscript, I have the following comments:
- The revised manuscript shows substantial improvement and addresses nearly all reviewer concerns.
- While the authors correctly note that no single internationally recognized standard for rosehip oil fatty acid composition exists, the response would be stronger if it included a critical synthesis of the available literature. Several published reports provide reference ranges for linoleic, α-linolenic, oleic, and minor fatty acids in rosehip oils from different regions. A short comparative discussion of how the current samples align or deviate from those reported ranges would give readers a clearer context for evaluating authenticity and quality.
- The resolution of Figure 1 is suboptimal. The spectral curves and axis labels appear pixelated and difficult to read at standard magnification. Please provide a higher-resolution version with sharper lines and clear labeling of axes, tick marks, and legends to ensure the data are interpretable. Vector-based formats (e.g., TIFF, EPS, or PDF) are recommended for clarity. Same issue with figure 3 as well.
Author Response
After reviewing the revised version of the manuscript, I have the following comments:
- The revised manuscript shows substantial improvement and addresses nearly all reviewer concerns.
- While the authors correctly note that no single internationally recognized standard for rosehip oil fatty acid composition exists, the response would be stronger if it included a critical synthesis of the available literature. Several published reports provide reference ranges for linoleic, α-linolenic, oleic, and minor fatty acids in rosehip oils from different regions. A short comparative discussion of how the current samples align or deviate from those reported ranges would give readers a clearer context for evaluating authenticity and quality.
Response 1: The authors have added a new Table 4, providing a comparison of the samples with rosehip oils from around the world.
- The resolution of Figure 1 is suboptimal. The spectral curves and axis labels appear pixelated and difficult to read at standard magnification. Please provide a higher-resolution version with sharper lines and clear labeling of axes, tick marks, and legends to ensure the data are interpretable. Vector-based formats (e.g., TIFF, EPS, or PDF) are recommended for clarity. Same issue with figure 3 as well.
Response 2. The figures have been reworked.